# Performance of highly sensitive and conventional rapid diagnostic tests for clinical and subclinical *Plasmodium falciparum* infections, and *hrp2/3* deletion status in Burundi

**David Niyukuri[1,2], Denis Sinzinkayo[1,3], Emma V. Troth[4], Colins O. Oduma[5], Mediatrice Barengayabo[1], Mireille Ndereyimana[1], Aurel Holzschuh[4], Claudia A. Vera-Arias[4], Yilekal Gebre[4], Kingsley Badu[6], Joseph Nyandwi[1,7], Dismas Baza[8], Elizabeth Juma[9], Cristian Koepfli[4]***

1 Doctoral School, University of Burundi, Bujumbura, Burundi, 2 South African DSI-NRF Centre of Excellence in Epidemiological Modelling and Analysis, Stellenbosch University, Stellenbosch, South Africa, 3 National Malaria Control Program, Bujumbura, Burundi, 4 Eck Institute for Global Health and Department of Biological Sciences, University of Notre Dame, Notre Dame, Indiana, United States of America, 5 Kenya Medical Research Institute, Kisumu, Kenya, 6 Kwame Nkrumah University of Science and Technology, Kumasi, Ghana, 7 National Institute of Public Health, Bujumbura, Burundi, 8 WHO Country Office, Bujumbura, Burundi, 9 WHO African Region, Accra, Ghana

* ckoepfli@nd.edu

**Data Availability Statement:** All data can be found in the supplementary file.

## Abstract

Rapid diagnostic tests (RDTs) are a key tool for the diagnosis of malaria infections among clinical and subclinical individuals. Low-density infections, and deletions of the *P. falciparum* *hrp2/3* genes (encoding the HRP2 and HRP3 proteins detected by many RDTs) present challenges for RDT-based diagnosis. The novel Rapigen Biocredit three-band *Plasmodium falciparum* HRP2/LDH RDT was evaluated among 444 clinical and 468 subclinical individuals in a high transmission setting in Burundi. Results were compared to the AccessBio CareStart HRP2 RDT, and qPCR with a sensitivity of <0.3 parasites/μL blood. Sensitivity compared to qPCR among clinical patients for the Biocredit RDT was 79.9% (250/313, either of HRP2/LDH positive), compared to 73.2% (229/313) for CareStart (*P* = 0.048). Specificity of the Biocredit was 82.4% compared to 96.2% for CareStart. Among subclinical infections, sensitivity was 72.3% (162/224) compared to 58.5% (131/224) for CareStart (*P* = 0.003), and reached 88.3% (53/60) in children <15 years. Specificity was 84.4% for the Biocredit and 93.4% for the CareStart RDT. No (0/362) *hrp2* and 2/366 *hrp3* deletions were observed. In conclusion, the novel RDT showed improved sensitivity for the diagnosis of *P. falciparum*.

**Funding:** This work was supported by the National Institutes of Health grant R21AI137891 to CK. The funders had no role in study design, data analysis, decision to publish, or preparation of the manuscript.

**Competing interests:** The authors declare that no competing interests exist.

## Introduction

Accurate diagnosis and treatment of malaria infections is a key pillar for control. Over the last decade, rapid diagnostic tests (RDTs) have become widely used for diagnosis. RDTs are lateral flow devices that detect parasite-specific proteins by immunohistochemistry. No laboratory infrastructure is required for RDT use, they can be used with minimal training, and results are available within 10–20 minutes [1, 2]. The sensitivity of RDTs for the diagnosis of clinical cases is similar or better than field microscopy, the only other method for diagnosis in health centers [3, 4]. RDTs are used in many peripheral health centers where microscopy is not available, and by community health workers. In 2020, nearly 300 million RDTs were used by malaria control programs [5].

In addition to clinical infections diagnosed at health centers, subclinical infections present a major challenge for control. While not causing febrile illness, subclinical infections can result in more subtle health impacts, e.g. anemia [6]. Using different approaches, several studies found that over 50% of transmission originates from asymptomatic infections [7–9]. A study conducted in western Kenya estimated 95% of transmission stemming from subclinical carriers [10]. RDTs are the key diagnostic tool for strategies to target the subclinical reservoir, for example through active [11, 12] or reactive case detection [13, 14],

Among clinical and subclinical infections, a proportion of infections are below the limit of detection of RDTs and remain untreated. These untreated infections might result in extended periods of illness, and sustain onward transmission. More sensitive RDTs might be needed to achieve further gains in malaria control and elimination.

Apart from low parasite density, deletions of the *hrp2* and *hrp3* genes are a threat to diagnosis by RDT. The most sensitive RDTs for *P. falciparum* detect the HRP2 and HRP3 proteins. Deletions of these genes have been initially described in Peru [15] and later in multiple countries, and result in false-negative RDTs even when parasite density is high. In Africa, deletions are particularly prevalent around the Horn of Africa [16–18], and have been described in multiple other countries, e.g. the Democratic Republic of Congo [19], and Ghana [20]. Alternative diagnostic targets include Plasmodium Lactate Dehydrogenase (pLDH) or Aldolase. These proteins are essential and thus genes cannot be deleted, but sensitivity of diagnosis is lower compared to HRP2 [21]. The WHO recommends regular surveillance of *hrp2*/*hrp3* deletion to select the optimal tool for diagnosis [22].

The level of malaria transmission in Burundi is among the highest in the world [23, 24]. In contrast to the trend in many other countries in sub-Saharan Africa, the number of cases has increased in the last two decades [24]. The increase was observed despite an increase in the number of health centers offering diagnosis and treatment, and increase in testing, and roll-out of interventions such as bed nets and indoor residual spraying (IRS) [24]. To understand the benefit of a novel, highly sensitive RDT, suspected clinical cases presenting to a health center, and subclinical community members were tested by the novel RDT, an established RDT, and results were compared to highly sensitive qPCR. *P. falciparum* positive infections were typed for *hrp2* and *hrp3* deletions.

## Study population and methods

### Ethical approval

Informed written consent was collected from each individual, or, in the case of minors, from the legal guardian prior to sample collection. This study was approved by the Comité National d'Ethique pour la protection des êtres humains sujets de la recherche biomédicale et comportementale of Burundi (approval no. CNE/03/2021), and the University of Notre Dame Institutional Review Board (approval no. 21-02-6446).

### Rapid diagnostic tests

The Biocredit Malaria Ag Pf (pLDH/HRPII) RDT (lot no. H052A001DA), manufactured by Rapigen, contains two test bands, one for HRP2 and one for LDH, in addition to the control band. Plasmodium Lactate Dehydrogenase (pLDH) is an enzyme of the glycolytic pathway and essential for the parasite. Histidine Rich Protein 2 and 3 are highly expressed proteins. Their function is not well understood [2]. Deletions of the *hrp2* and *hrp3* genes result in false-negative RDTs. The test was evaluated previously among microscopy-positive individuals and showed very high agreement with diagnosis by microscopy [25]. It has not been not evaluated among clinical patients that might harbor submicroscopic infections, or subclinical individuals. Given the separated HRP2 and LDH bands, the test can be used for surveillance of possible *hrp2* deletion. Differences in sensitivity of HRP2 and LDH can be studied using a single test. This test meets critical WHO criteria and is currently undergoing WHO prequalification [26].

As comparison, the AccessBio CareStart Malaria Pf (HRP2) RDT was used (lot no. MO19H71). This HRP2-based RDT is WHO pre-qualified [26] and has been in use for approximately ten years [27–29]. It is the main RDT used by health systems in multiple African countries [20, 30].

Testing procedures are identical for both RDTs and are explained in detail e.g. in [31]. A droplet of blood (approximately 5 μL), collected either by finger prick or phlebotomy, is put onto a specific spot on the test kit, followed by 2–3 drops of assay diluent buffer on a separate spot. Results are read after 15–20 minutes. The cost for either test is approximately 1 USD.

### Sample collection and RDT diagnosis

Burundi experiences high, year-round transmission of *P. falciparum* malaria with moderate seasonality. Samples were collected in Cibitoke Province in northwestern Burundi. Cibitoke Province borders Rwanda and the Democratic Republic of Congo (DRC). Samples were collected during the dry, lower transmission season in June 2021. Clinical samples were collected in health centers. All patients presenting for malaria diagnosis were invited to join the study. Individuals found positive were treated according to national guidelines. Subclinical samples were collected on a market, with all individuals present invited to join the study.

Approximately 200 μL blood was collected by finger prick into EDTA tubes. Two RDTs were run on site. Remaining blood was placed into -20˚C storage every evening and kept at -20˚C until DNA extraction.

### *P. falciparum* qPCR and *hrp2/3* deletion typing

DNA was extracted using the Macherey-Nagel NucleoMag Blood 200 μL kit according to manufacturer's instructions. This kit yields very high DNA recovery, resulting in a low limit of detection [32]. DNA was extracted from 200 μL blood and eluted in 100 μL elution buffer. qPCR was done in a total volume of 12 μL, including 2 μL DNA, corresponding to 4 μL blood. The *P. falciparum varATS* assay was used. This assay targets a multicopy gene that is present in approximately 60 copies per parasite, of which approximately 20 copies are amplified with the primers and probe used. It thus offers very high sensitivity of <0.3 parasites/μL blood [33]. qPCR conditions are given in S1 File. For absolute quantification of parasite density, a standard curve derived from DNA from cultured 3D7 parasites and quantified by droplet digital PCR (ddPCR) was run along samples. Samples positive for *P. falciparum* at a density of approximately >5 parasites/μL were typed for *hrp2* and *hrp3* deletion by ddPCR [34]. In this assay, *hrp2* or *hrp3* and a control gene (*serine-tRNA ligase*) are quantified in a single tube with very high specificity, thus providing highly accurate data on deletion status [34]. The assay amplifies parts of *hrp2* exon 2 that encodes for the antigen detected by the RDT. The amplified

region is deleted in all reported cases where *hrp2* deletion breakpoints were studied [35]. Assay conditions are given in S1 File.

## Data analysis

The sample size was determined to be able to detect a 5% difference in sensitivity with 95% confidence. In the absence of any preliminary data, a prevalence of infection of 50% was assumed, yielding the largest sample size of n = 377. The number of individuals sampled (n = 444 clinical and n = 468 subclinical) was above the minimum target, these numbers were reached out of convenience.

The following performance characteristics were determined for both RDTs, and the HRP2 and LDH bands on the Biocredit RDT separately as well as in combination: (i) sensitivity: the number of infections detected by an RDT divided by the total number of infections detected by qPCR as gold standard. (ii) Specificity: the proportion of negative RDTs among individuals that tested negative by qPCR. (iii) Limit of detection (LoD): LoD was defined as the lowest parasite density where a qPCR-positive infection would be detected with 95% probability. To determine the limit of detection (LoD) of each RDT target, logistic regression analysis was conducted. (iv) Area under the receiver operating characteristic curve (ROC) curve (AUC): AUC was calculated with a nonparametric analysis using 1000 bootstrap replications, with log10 transformed parasite density by qPCR as classification variable. Diagnostic accuracy was considered excellent if AUC was >0.9, and very good for AUC values of >0.8 to <0.9 [36].

As parasite density distributions were skewed, geometric mean densities are given whenever densities are reported. CI95 stands for 95% confidence interval. All data can be found in S1 Data.

## Results

### Study population demographics

Demographic data of study participants are given in Table 1. Due to the convenience sampling strategy, the subclinical study population does not represent the general population. Only 27/468 subclinical individuals were children below 5 years of age, and only 73/468 were children aged 5–15 years. The majority of those sampled were female.

Table 1. Demographics of study population and prevalence and test positivity (by qPCR and RDT) by age group and gender.

| Clinical | | | N | Test positivity by qPCR | *P* | Test positivity by RDT | *P* | RDT sensitivity | *P* |
|---|---|---|---|---|---|---|---|---|---|
| Age group | 0–5 | | 158 (35.6%) | 62.0% | | 49.4% | | 72.4% | |
| | 5–15 | | 99 (22.3%) | 82.8% | 0.002 | 86.9% | <0.001 | 96.3% | <0.001 |
| | >15 | | 187 (42.1%) | 71.1% | | 58.3% | | 75.2% | |
| Gender | Male | | 181 (59.2%) | 67.4% | | 62.7% | | 82.0% | |
| | Female | | 263 (40.8%) | 72.6% | 0.236 | 59.7% | 0.514 | 78.5% | 0.460 |
| **Subclinical** | | | N | Prevalence by qPCR | *P* | Prevalence by RDT | *P* | RDT sensitivity | *P* |
| Age group | 0–5 | | 27 (5.8%) | 51.8% | | 55.6% | | 78.6% | |
| | 5–15 | | 73 (15.6%) | 63.0% | 0.015 | 67.1% | <0.001 | 91.3% | 0.003 |
| | >15 | | 368 (78.6%) | 44.7% | | 36.8% | | 66.5% | |
| Gender | Male | | 217 (46.5%) | 50.7% | | 39.6% | | 75.5% | |
| | Female | | 250 (53.5%) | 45.6% | 0.272 | 46.1% | 0.158 | 69.3% | 0.303 |

For RDT data, results from the Biocredit test were included, with either band (HRP2 or LDH) positive counting as a positive test. Test positivity and prevalence by RDT includes individuals that were RDT positive but qPCR negative, resulting in higher test positivity or prevalence by RDT than qPCR in some groups. RDT sensitivity is calculated as proportion of RDT positive individuals among those positive by qPCR. *P*-values to compare groups were calculated by Chi-square test.

## RDT sensitivity and specificity

444 clinical patients presenting to a health center were enrolled. By qPCR, *P. falciparum* positivity was 70.5% (313/444). Mean parasite density by qPCR was 727.4 parasites/μL (CI95: 405.9, 1303.6). Counting either band (HRP2 or LDH) positive, the sensitivity of the Biocredit RDT was 79.9% compared to qPCR (250/313 of infections detected, Fig 1A, Tables 2 and 3). 23 individuals tested positive by RDT, but negative by qPCR, resulting in a specificity of 82.4%.

Among 468 subclinical individuals enrolled, 47.9% (224/468) were positive by qPCR. Mean parasite density was 22.9 parasites/μL (CI95: 13.1, 40.3). The sensitivity of the Biocredit test was 72.3% compared to qPCR (162/224, Fig 1A, Tables 2 and 3). 38 individuals were positive by RDT but negative by qPCR, resulting in a specificity of 84.4% (194/230).

The novel Biocredit RDT was compared against the CareStart RDT, an established product routinely used in multiple countries (Fig 1A, Table 3). The CareStart RDT showed a sensitivity of 73.2% (229/313) compared to qPCR among clinical individuals and thus was slightly less sensitive than the Biocredit RDT with a sensitivity of 79.9% (*P*<0.001, Table 2). Five individuals tested positive by CareStartRDT but negative by qPCR, resulting in a specificity of 96.2%. Among subclinical individuals, sensitivity of the CareStart was 58.5% (131/224), and thus substantially lower than the Biocredit at 72.3%. Specificity was 93.4% (228/244, *P*<0.001, Table 2).

On the Biocredit RDT, the HRP2 band was more sensitive than the LDH band (Fig 1B). Among clinical individuals, sensitivity of HRP2 was 77.6% (243/313), compared to 71.9%

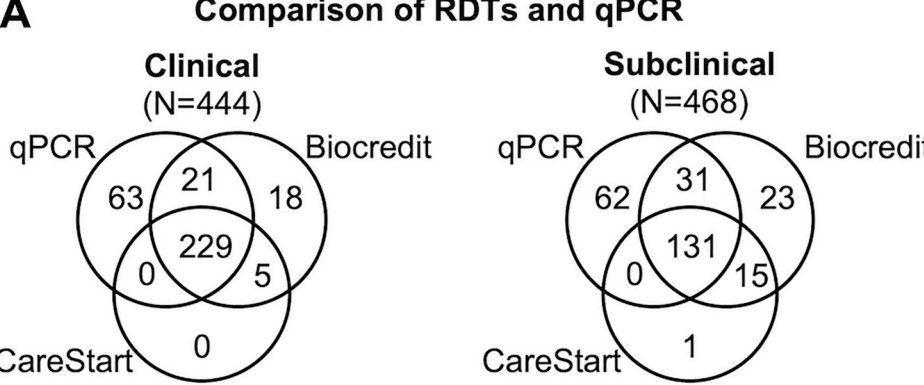

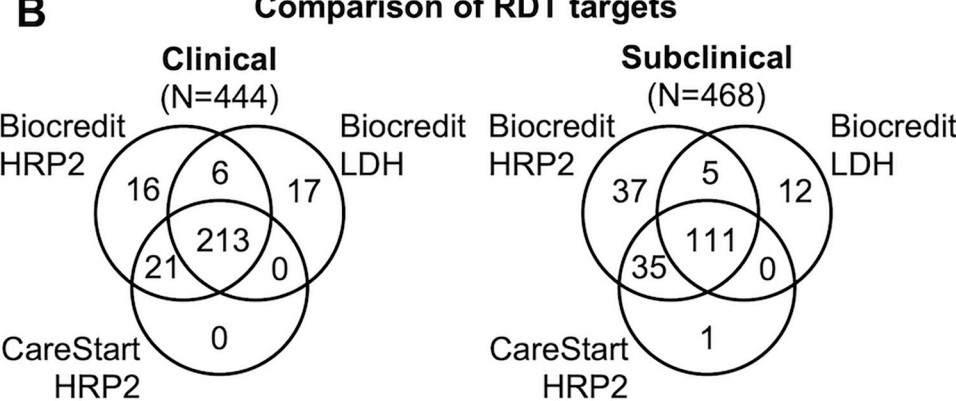

**Fig 1. Correspondence among qPCR, and different RDTs and RDT targets.** For panel A, when either band on the Biocredit RDT was positive it was recorded as positive. Panel B includes false-positive RDTs that were negative by qPCR.

**Table 2. Limit of detection, AUC, and sensitivity and specificity of RDTs as compared to qPCR.**

| | LoD [parasites/μL] [CI95] | Clinical | | | Subclinical | | |
|---|---|---|---|---|---|---|---|
| | | Sensitivity [CI95] | Specificity [CI95] | AUC [CI95] | Sensitivity [CI95] | Specificity [CI95] | AUC [CI95] |
| Biocredit any band | 34 [17, 82] | 79.9% [75.0, 84.0] | 82.4% [74.9, 88.1] | 0.939 [0.913, 0.964] | 72.3% [66.1, 77.8] | 84.4% [79.3, 88.5] | 0.862 [0.811, 0.913] |
| Biocredit HRP2 | 42 [22, 96] | 77.6% [72.7, 81.9] | 90.1% [83.6, 94.2] | 0.955 [0.935, 0.975] | 71.0% [64.7, 76.6] | 88.1% [83.4, 91.6] | 0.863 [0.813, 0.914] |
| Biocredit LDH | 290 [159, 617] | 71.9% [66.6, 76.6] | 91.6% [85.4, 95.3] | 0.965 [0.945, 0.984] | 50.0% [43.5, 56.5] | 93.4% [89.6, 96.0] | 0.907 [0.869, 0.944] |
| CareStart HRP2 | 178 [94, 399] | 73.2% [68.0, 77.8] | 96.2% [91.1, 98.4] | 0.962 [0.943, 0.981] | 58.5% [51.9, 64.8] | 93.4% [89.5, 96.0] | 0.867 [0.818, 0.916] |

(225/313) for the LDH band (*P*<0.001, Table 2). Specificity was 90.1% (118/131) for HRP2, and 91.6% (120/131) for LDH (Table 2). Among subclinical individuals, the difference in sensitivity between HRP2 and LDH was more pronounced than among clinical infections. Sensitivity of HRP2 was 71.0% (159/224), compared to 50.0% (112/224) for LDH (*P*<0.001, Table 2). Specificity for HRP2 only was 88.1% (215/244), and for LDH only it was 93.4% (214/230, Table 2). Low correspondence between false-positive HRP2 and LDH bands was observed. Among 61 false-positive RDTs (clinical and subclinical combined), only 8 were false-positive for both targets, 34 were positive for HRP2 only, and 19 for LDH only.

The LoD, defined as the lowest parasite density that could be detected with 95% probability, was 34 parasites/μL (CI95 17, 82) for the Biocredit RDT (any band), 42 parasites/μL (CI95 22, 96) for the Biocredit HRP2 band, 290 parasites/μL (CI95 159, 617) for the Biocredit LDH band, and 178 parasites/μL (CI95 94, 399) for CareStart (Table 2, Fig 2).

The diagnostic accuracy as measured by AUC analysis was excellent (>0.9) among clinical samples for both RDTs and each target on the Biocredit RDT (Table 2). Among subclinical individuals, diagnostic accuracy was very high (>0.8) for all RDTs, and reached 0.907 for the LDH band on the Biocredit.

## *hrp2/3* deletion typing

362 *P. falciparum* positive samples were successfully typed for *hrp2* deletion, and 366 were successfully typed for *hrp3* deletion. No *hrp2* deletions were observed. Two samples carried *hrp3*

**Table 3. Number of positives by RDT vs. qPCR, and by Biocredit vs. CareStart.**

| A) Clinical infections | | | | B) Subclinical infections | | | |
|---|---|---|---|---|---|---|---|
| | | Biocredit (any band) | | | | Biocredit (any band) | |
| | | Negative | Positive | | | Negative | Positive |
| qPCR | Negative | 108 | 23 | qPCR | Negative | 206 | 38 |
| | Positive | 63 | 250 | | Positive | 62 | 162 |
| | | Carestart | | | | Carestart | |
| | | Negative | Positive | | | Negative | Positive |
| qPCR | Negative | 126 | 5 | qPCR | Negative | 228 | 16 |
| | Positive | 84 | 229 | | Positive | 93 | 131 |
| | | Biocredit (any band) | | | | Biocredit (any band) | |
| | | Negative | Positive | | | Negative | Positive |
| Carestart | Negative | 171 | 39 | Carestart | Negative | 267 | 54 |
| | Positive | 0 | 234 | | Positive | 1 | 146 |

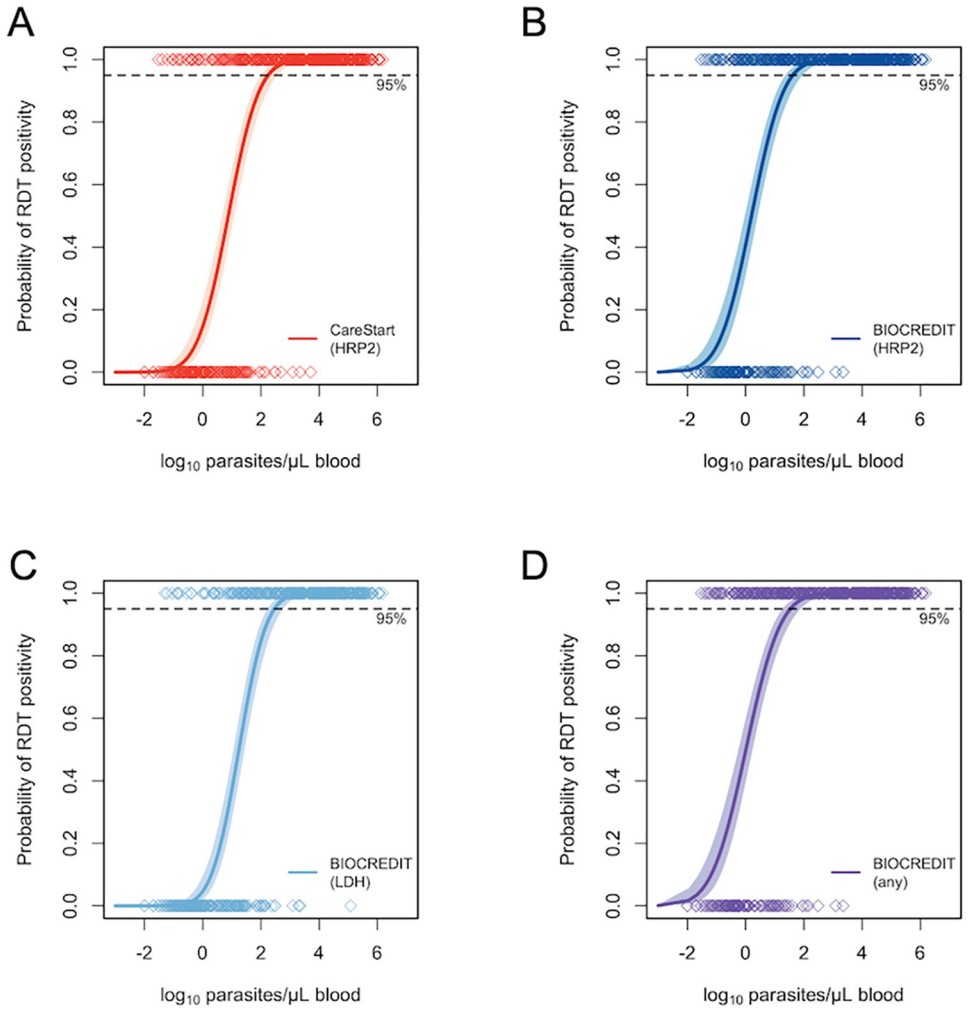

**Fig 2. Limit of Detection (LoD) of each RDT and RDT target in relation to parasite density.** Parasite density (by qPCR) of RDT positive and RDT negative samples is shown along the top and bottom X-axis. Shaded lines show 95% confidence intervals. A) Biocredit (any band) B) Biocredit HRP2 C) Biocredit LDH D) CareStart HRP2.

deletion, one each collected from a clinical and subclinical patient. The subclinical individual was negative by all RDTs at a density of 39 parasites/μL by ddPCR. The clinical individual was RDT positive at a density of 136,000 parasites/μL.

## Demographic risk factors for infection

Prevalence of subclinical infection, and test positivity rate peaked in children aged 5 to 15 years, both by qPCR and RDT (Table 1). Parasite densities decreased with age in clinical and subclinical infections (clinical: 9.7% decrease per year, $P<0.001$; subclinical: 4.6% decrease per year, $P = 0.003$). As a result, RDT sensitivity tended to be higher in children than adults. Among clinical infections, sensitivity of the Biocredit RDT (any band) was 83.3% (150/180) in children below 15 years compared to 75.2% (100/133) in adults aged 15 and older ($P = 0.076$). Among subclinical infections, sensitivity was 88.3% (53/60) in children and 66.5% (109/164) in adults ($P = 0.001$).

No significant differences were observed between men and women in prevalence by qPCR or RDT, or in RDT sensitivity (Table 1).

## Discussion

This study found high sensitivity of a novel RDT for the diagnosis of *P. falciparum* infections in clinical and subclinical individuals in highly endemic setting in Burundi. Using a highly sensitive qPCR with a limit of detection of <0.3 parasites/µL blood as gold standard, the RDT detected 80% of clinical and 72% of subclinical infections. The novel Biocredit RDT performed substantially better compared to the established CareStart RDT that is used by malaria control programs in multiple countries.

Due to the acquisition of natural immunity after repeated infections, parasite densities in children are highest, they are at highest risk of severe malaria, and in high transmission settings they are an important reservoir of transmission. The convenience sampling on a market resulted in an underrepresentation of children among subclinical individuals, with only 21% (100/468) study participants being below 15 years of age, compared to approximately 45% in the general population [37]. In the small group of 60 children that were positive by qPCR, the Biocredit RDT detected 88% of infections. Testing a group with an age distribution more representative of the general population, it is expected that the overall sensitivity of the RDT would be even higher than the 72% measured in the present study.

Compared to the established CareStart RDT, the LoD (defined as the parasite density where the probability of a positive RDT is 95%) of the Biocredit RDT was 5-fold lower. The LoD of the Biocredit HRP2-band only was 4-fold lower compared to the CareStart RDT, and the LoD of the Biocredit LDH-band only was moderately higher than the one of the CareStart. Of note, differences in the protocols used for DNA extraction [32], and in the external standards used for qPCR [38], result in pronounced differences in parasite quantification. Results from different studies are thus not directly comparable. We extracted DNA from whole blood and used an extraction protocol that yielded fourfold more DNA than another common protocol. Further, extraction from whole blood yielded tenfold more DNA than from DBS [32]. Using these protocols resulted in high DNA recovery, and thus in higher LoDs.

No *hrp2* deletions were detected in the study population. Differences in the number of tests positive by LDH and HRP2 are thus fully caused by differences in the LoD of these targets. On the Biocredit RDT, HRP2 was substantially more sensitive than LDH. Where no deletions are present, HRP2-based RDTs remain the most sensitive diagnostic target for RDTs. Among clinical infections, sensitivity of the Biocredit LDH band was very similar to the AccessBio HRP2-based RDT (72% vs. 73%). Countries currently using the Carestart HRP2 RDT that switch to non-HRP2-based RDTs due to high levels of deletion can expect similar sensitivity of diagnosis in clinical settings using the new test as with previous HRP2-based RDTs.

A high rate of false-positive RDTs was observed, and consequently a relatively low specificity. Antigens can persist after parasite clearance [39], though LDH levels were shown to follow parasite density closely [39]. Antigens persisting after treatment were shown to be a major cause of false-positive RDTs [40]. Recent treatment was not assessed in this study, thus this factor could not be assessed as a cause for false-positive RDTs. False-positive RDTs can also be the result of non-*Plasmodium* infections [41, 42] and non-infectious diseases [43]. Most false-positive Biocredit RDTs were positive by one target only, either HRP2 or LDH. Using an RDT with two bands reduces specificity compared to a single-target test.

Half of the mostly adult study population in the study site in Burundi's north-west carried subclinical *P. falciparum* infections. These individuals are expected to be an important source of transmission [7–10]. Interventions targeting the asymptomatic reservoir might be needed to reduce transmission. The high sensitivity of the Biocredit RDT to diagnose subclinical infections opens up new avenues for malaria control, e.g. through mass screen and treat campaigns.

In conclusion, the Biocredit RDT offers high sensitivity for the diagnosis of clinical and sub-clinical infections. Parasite density distributions differ across transmission intensities, and more subpatent infections are generally observed as transmission declines [44, 45]. Studies in regions of medium and low transmission will be required to determine sensitivity and specificity in these populations. Given the absence of *hrp2* deletions, HRP2-based diagnostic remains appropriate in the site of the current study. Surveillance for *hrp2* and *hrp3* deletions will be required in other sites across Burundi.

## Supporting information

**S1 File. Laboratory protocols: *P. falciparum var*ATS qPCR, *hrp2* deletion, *hrp3* deletion.** (DOCX)

**S1 Data. Database.** (TXT)

## Acknowledgments

We thank all study participants and field teams.

## Author Contributions

**Conceptualization:** Kingsley Badu, Cristian Koepfli.

**Data curation:** Cristian Koepfli.

**Formal analysis:** Aurel Holzschuh, Cristian Koepfli.

**Funding acquisition:** David Niyukuri, Cristian Koepfli.

**Investigation:** David Niyukuri, Denis Sinzinkayo, Emma V. Troth, Colins O. Oduma, Mediatrice Barengayabo, Mireille Ndereyimana, Claudia A. Vera-Arias, Yilekal Gebre.

**Project administration:** David Niyukuri, Mediatrice Barengayabo, Cristian Koepfli.

**Resources:** Joseph Nyandwi, Dismas Baza, Elizabeth Juma, Cristian Koepfli.

**Supervision:** David Niyukuri, Colins O. Oduma, Cristian Koepfli.

**Validation:** Cristian Koepfli.

**Writing – original draft:** Cristian Koepfli.

**Writing – review & editing:** David Niyukuri, Denis Sinzinkayo, Emma V. Troth, Colins O. Oduma, Claudia A. Vera-Arias, Kingsley Badu, Cristian Koepfli.

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
