## [Decision Letter · Decision Letter 0]

2 Jun 2022

PGPH-D-22-00743

A novel rapid diagnostic test for highly sensitive diagnosis of clinical and subclinical Plasmodium falciparum infections

Dear Dr. Koepfli,

Thank you for submitting your manuscript to PLOS Global Public Health. After careful consideration, we feel that it has merit but does not fully meet PLOS Global Public Health’s publication criteria as it currently stands. Therefore, we invite you to submit a revised version of the manuscript that addresses the points raised during the review process.

Specific items raised by the reviewers that will need to be addressed before the manuscript can proceed for publication are:

Reviewers 1 and 2 raise the need for more detail and clarification on the statistical methods used for evaluating the performance of the RDTs. On this item, reviewer 2 raises concern about why the area under the ROC curve was not calculated.Reviewer 2 raises concern about the validity of the conclusions drawn on the prevalence of sub-clinical malaria within the local population.Reviewers 1 and 3 raise the need for more information on the RDT used, including the Lot number, and clarification on whether details on this RDT have been published previously.As detailed by reviewer 1, further details and clarification are needed on several laboratory methods.

I recommend addressing the other items raised by the reviewers to improve the quality of the manuscript.

Please submit your revised manuscript by . If you will need more time than this to complete your revisions, please reply to this message or contact the journal office at globalpubhealth@plos.org. Please include the following items when submitting your revised manuscript:

We look forward to receiving your revised manuscript.

Kind regards,

Sarah Auburn

Academic Editor

Journal Requirements:

State the initials, alongside each funding source, of each author to receive each grant.

2. Please provide separate figure files in .tif or .eps format only and remove any figures embedded in your manuscript file. Please also ensure that all files are under our size limit of 10MB.

Additional Editor Comments (if provided):

Reviewers' comments:

Reviewer's Responses to Questions

**Comments to the Author**

1. Does this manuscript meet PLOS Global Public Health’s publication criteria? Is the manuscript technically sound, and do the data support the conclusions? The manuscript must describe methodologically and ethically rigorous research with conclusions that are appropriately drawn based on the data presented.

Reviewer #1: Yes

Reviewer #2: Yes

Reviewer #3: Yes

2. Has the statistical analysis been performed appropriately and rigorously?

Reviewer #1: I don't know

Reviewer #2: No

Reviewer #3: Yes

3. Have the authors made all data underlying the findings in their manuscript fully available (please refer to the Data Availability Statement at the start of the manuscript PDF file)?

Reviewer #1: No

Reviewer #2: Yes

Reviewer #3: Yes

4. Is the manuscript presented in an intelligible fashion and written in standard English?

Reviewer #1: No

Reviewer #2: Yes

Reviewer #3: Yes

5. Review Comments to the Author

Reviewer #1: Title: A novel rapid diagnostic test for highly sensitive diagnosis of clinical and subclinical Plasmodium falciparum infections

Authors: Niyukuri et al.

Comments

Abstract: Although the results (sensitivity) of the novel Rapigen Biocredit three-band Plasmodium falciparum HRP2/LDH RDT for clinical (febrile) and subclinical individuals were compared against AccessBio CareStart HRP2 RDT, the result doesn’t include the performance of novel Rapigen Biocredit three-band Plasmodium falciparum HRP2/LDH RDT performance against qPCR in the abstract section.

The terms clinical and febrile seem interchangeably used. It would be nice if authors consistently use either in the text.

There is no mention on the recruitment processes of 444 clinical and 468 subclinical individuals. What specific tools were used to diagnose malaria at those health facilities? Were treatments given to those tested positive for malaria?

Line 14, ………… present challenges for RDT-based diagnosis. This should read as ‘….present challenges to HRP2/3-based RDTs.

Introduction

It is well written.

MM

Rapid diagnostic tests: the description presented under this subheading shall be written under the introduction section. This section perhaps should describe how the authors have conducted the evaluation instead of narrating the use of different RDTs and the significance of the different target antigens to the parasite. The lot number and other important parameters of the RDTs are not well stated.

Authors shall report the situation of malaria (including species composition) in the study area so that endemicity of the area can be better understood. Indeed, the fact that market places were used for the recruitment of subclinical individuals, there was no justification for choosing such places although there is a mention on using convenience sampling strategy.

Sample collection and RDT diagnosis

The clinical samples were collected in self-reported febrile patients at health-facilities; however, the subclinical individuals were recruited from market. Why the market place is chosen?

Line 108/109: …….’Only 27/468 subclinical individuals were children below 5 years of age, and only 73/468 were children aged 5- 109 15 years’. The majority of those sampled were female. These sentences are wrongly placed under this section (MM) as they describe results and should be moved to the result section. Moreover, Table 1 also describes results and hence shall move to the appropriate place, result. Caption should be written above the table.

This section is haphazardly written and shall be revised accordingly.

Line 128: P. falciparum qPCR and hrp2/3 deletion typing

Authors shall describe how they collected blood samples before embarking on the procedures used for DNA extraction, qPCR and hrp2/3 gene del. What is the expected bp of hrp2 exon2?

Did authors use microscopy for the detection and quantification of parasitaemia?. Details are needed.

Details of the primers used for qPCR and the cycling conditions are not described.

Details on how authors did hrp2/3 gene deletion are needed. Did authors amplify exon 2, 3 including the upstream and downstream flanking regions? Assessment of hrp2/3 deletion should encompass the entire coding region. ddPCR is inadequately presented.

How the sample size (444 clinical and 468 subclinical) was calculated?

Results

In Table 1 (line 117-118), the test positivity by qPCR, novel RDT, the conventional RDT (CareStart RDT) shall be reported in such a way that the results could be easily accessed.

It is unclear why authors did not use microscopy as part of the comparison of the different malaria diagnostic tools/

Table 2 (155-156): the LoD (parasites/uL) for the different types of RDTs presented in the table is not well reviewed in the background information. References are needed as far as LoD is concerned for those RDTs. The caption of table two needs revision to conform to the message it conveys (clinical and subclinical samples).

The calculation of sensitivity and specificity of the RDTs should be compared against the qPCR, so that the activity of each of the RDT can be appreciated.

Line 164-165: it reads ‘the sensitivity of the Biocredit test was 72.3% (162/238, Figure 165 1A, Table 2)’, however, this value is incorrect since 162/238 gives 68.1%, but not 72.3%. Instead, the sensitivity of the Biocredit could be 162/224.

The sensitivity and specificity of the Biocredit can be calculated using 2 x 2 tables, so that the results of the Biocredit against qPCR can easily be appreciated.

Line 168-169: I am not sure whether authors used novel Biocredit RDT or CareStart RDT as a reference while calculating sensitivity. The sensitivity of the Biocredit RDT and CareStart RDT shall be compared against the reference test (qPCR).

The results displayed in Fig.2 are not readable and needs revision.

Discussion

It is interesting to note that qPCR detected falciparum malaria in 70.5% (313/444) of the clinical samples and 47.9% (224/468) malaria in the subclinical samples. Is there any data on the prevalence of subclinical malaria in the study area before?

In line 218-219, authors report that the novel RDT detected 80% of clinical and 72% of the subclinical infections. What about CareStart RDT? The prevalence of clinical and subclinical is comparable and this needs strong justifications to be valid.

Authors did not discuss the likely reasons for detecting 82.8% of the clinical and 63% of subclinical samples were tested qPCR positive among 5-15 years.

Reviewer #2: Review of the manuscript: “A novel rapid diagnostic test for highly sensitive diagnosis of clinical and subclinical Plasmodium falciparum infections” (PGPH-D-22-00743)

Summary: This article assessed the performance of a novel P. falciparum specific lateral flow assay against a commercially available rapid diagnostic test and qPCR with a theoretical LOD of <0.1 parasites / µl. Through non-random selection procedures a total of 444 clinical Pf patients and 468 asymptomatic participants were enrolled. Sensitivity of the novel diagnostic against qPCR was 80% and 73% for clinical and asymptomatic participants respectively. The novel diagnostic showed better performance than the commercially available RDT>

Major Comment: This is a well written article covering a topic of public health. My main concern is that the authors did not calculate the area under the ROC curve, which would have made it much easier to compare performance of both RDTs directly as well as both antigens.

In addition, the statistical methods as stated right now are not sufficient. In case the authors do not want to compare the areas under the ROC, they at least need to clarify in the methods what approach they have used to compare performance?

Finally the authors draw conclusions about the prevalence of sub-clinical malaria within the local population (in lines 256 and thereafter), considering the sampling method such a conclusion cannot be made.

Other comments:

• Can you provide some background on the novel RDT in the introduction? For example:

o Test procedures

o Time to result

o Costs

• Table 1 contains a large proportion of all results, however is located within the methods section, consider moving this table to the results section?

Minor Comments:

• Kindly include assay specific specificity in the abstract as well

• In line 42 the authors suggest that sensitivity of RDTs is superior to light microscopy and it would be good if this statement was backed up by a reference.

• Line 68: this sentence needs to be revised for grammar

• Lines 72 – 75: this section appears more suitable with the methods section?

• Line 97: can you clarify if the Carestart RDT is WHO pre-qualified – I think it is?

• Lines 107 – 109 are results and seem better placed in the results than methods section

• Line 126: How long did it take on average before samples were stored at -20°C?

• Line 144: there is a funny sentence around CI, which should probably not be there?

• Kindly extend the methods and describe briefly how performance was calculated

• Table 2: add 95% confidence intervals to performance indicators

• Lines 219 – 220: please use the full name of either diagnostic rather than talking about an “established test”

• Lines 244 – 245 states that countries switching from HRP2 based RDTs to non-HRP2 diagnostics can expect a comparable performance. However, from the presented data it appears that sensitivity dropped from 77% to 72%, which seems like quite a lot. Kindly use a sloid statistical method to determine whether this difference is significant and discuss accordingly

• Line 250: clarify that by antigen you mean HRP2

• Line 256 states that half the population carried sub-clinical Pf infections. Given the non-random nature of sampling this statement cannot be made, kindly remove.

Reviewer #3: The manuscript presented by the authors contains interesting and important information for malaria research community and also for the malaria control programs that are looking for new/better diagnostic tools. However there are some comments to the authors and some corrections that need to be addressed before the publication:

1. The title mention "A novel rapid diagnostic test..."; but looking at the literature there is a paper from 2020 where the authors compare 3 commercial RDTs from BIOCREDIT, including the Pf (HRP2/LDH). This paper, by Seo Hye Park et al. (Korean J Parasitol. 2020 Apr; 58(2): 147–152. Published online 2020 Apr 30. doi: 10.3347/kjp.2020.58.2.147), should be included in the discussion also. In addition, these tests were also part of the Malaria Rapid Diagnostic Test Performance-Results of WHO product testing of malaria RDTs: round 6 (2014-2015).

Could the authors please confirm if the test they used are the same (just different lot) or is it a new (improved) version?, if so, please give more details about it and also include lot number.

2. In the summary (line 31) it is mention ..."and reached 88.3% (53/60) in children < 15 years". By other hand in the results section (line 207) it says..."sensitivity was 88.3% (53/61) in children"...

Please correct the number: 60 or 61 children < 15 years?.

3. Table 1 is named "Demographics of study population and risk factors for infection (by qPCR and RDT)"...The percentage of each age group should be included and also the percentage of male and female. In addition, it is not explain how do the did the analysis of risk factors. Finally, if the authors want to keep "test positivity and prevalence by qPCR and RDT", I suggest to include also the data on specificity.

4. This paper also includes data on hrp2/3 deletion typing, an important issue in African endemic countries, so this should be mention in the objectives and, if possible, also in the title.

5. In the conclusion, in addition to mention the high sensitivity for the diagnosis of clinical and subclinical infections, it should also mention the absent of hrp2 deletion. The rest of the text in this paragraph (lines 265-267) could be merged with the previous paragraph (lines 256-262) as part of the discussion and future studies.

6. PLOS authors have the option to publish the peer review history of their article (what does this mean?). If published, this will include your full peer review and any attached files.

**Do you want your identity to be public for this peer review?** For information about this choice, including consent withdrawal, please see our Privacy Policy.

Reviewer #1: **Yes: **Lemu Golassa

Reviewer #2: No

Reviewer #3: No

---

## [Decision Letter · Decision Letter 1]

6 Jul 2022

Performance of highly sensitive and conventional rapid diagnostic tests for clinical and subclinical Plasmodium falciparum infections, and hrp2/3 deletion status in Burundi

PGPH-D-22-00743R1

Dear Mr. Koepfli,

We are pleased to inform you that your manuscript 'Performance of highly sensitive and conventional rapid diagnostic tests for clinical and subclinical Plasmodium falciparum infections, and hrp2/3 deletion status in Burundi' has been provisionally accepted for publication in PLOS Global Public Health.

Best regards,

Sarah Auburn

Academic Editor

Reviewer Comments (if any, and for reference):

Reviewer's Responses to Questions

**Comments to the Author**

1. If the authors have adequately addressed your comments raised in a previous round of review and you feel that this manuscript is now acceptable for publication, you may indicate that here to bypass the “Comments to the Author” section, enter your conflict of interest statement in the “Confidential to Editor” section, and submit your "Accept" recommendation.

Reviewer #1: (No Response)

Reviewer #2: All comments have been addressed

2. Does this manuscript meet PLOS Global Public Health’s publication criteria? Is the manuscript technically sound, and do the data support the conclusions? The manuscript must describe methodologically and ethically rigorous research with conclusions that are appropriately drawn based on the data presented.

Reviewer #1: Yes

Reviewer #2: Yes

3. Has the statistical analysis been performed appropriately and rigorously?

Reviewer #1: Yes

Reviewer #2: Yes

4. Have the authors made all data underlying the findings in their manuscript fully available (please refer to the Data Availability Statement at the start of the manuscript PDF file)?

Reviewer #1: Yes

Reviewer #2: Yes

5. Is the manuscript presented in an intelligible fashion and written in standard English?

Reviewer #1: Yes

Reviewer #2: Yes

6. Review Comments to the Author

Reviewer #1: (No Response)

Reviewer #2: Many thanks for considering my comments. There is only one point I would like to raise again: in line 173 - 174 of the clean manuscript you have revised a sentence which now reads: "Due to the convenience sampling strategy, the subclinical study population does not fully represent the general population". Kindly consider to remove the "fully" since a population either is or is not representative and the study population enrolled in this study is not reflective of the local population. Many thanks.

7. PLOS authors have the option to publish the peer review history of their article (what does this mean?). If published, this will include your full peer review and any attached files.

**Do you want your identity to be public for this peer review?** For information about this choice, including consent withdrawal, please see our Privacy Policy.

Reviewer #1: No

Reviewer #2: **Yes: **Benedikt Ley
